# Using Intermediate Forward Iterates for Intermediate Generator Optimization

### Abstract

Score-based models have become increasingly popular for image and video generation. In score-based models, a generative task is formulated using a parametric model (such as a neural network) to directly learn the gradient of such high dimensional distributions, instead of the density functions themselves, as is done traditionally. The key advantage in this framework is that gradient information from trained models can be utilized in reverse by stochastic sampling to generate diverse samples. However, from a computational perspective, existing score-based models can be efficiently trained only if the forward or the corruption process can be computed in closed form. By using the relationship between the process and layers in a feed-forward network, we derive a backpropagation-based procedure, which we call *Intermediate Generator Optimization*, to utilize intermediate iterates of computable corruption process with marginal overhead in terms of memory. The main advantage of IGO is that it can be incorporated into any standard autoencoder pipeline for generative tasks. We analyze the sample complexity properties of IGO to solve Generative PCA as a downstream task. We apply IGO empirically on two dense predictive tasks, viz., image extrapolation, and point cloud denoising. Our experiments indicate that it is possible to obtain an ensemble of generators for various time points using first-order methods.

## 1   Introduction

Generative Adversarial Networks (GANs) were first shown to be successful in generating high-resolution realistic natural images Wu et al. (2019), and biomedical images Harutyunyan et al. (2001) for augmentation purposes. In image generation, Variational Autoencoders (VAEs) is a popular alternative which is based on minimizing the *distortion* given by integral metrics such as KL divergence. In both GANs and VAEs, the learning problem coincides. We seek to learn the process of generating new samples based on latent space modeled as random distributions Razavi et al. (2019); Zhang et al. (2019). Some of the applications enabled by such deep generative models (DGMs) in computer vision include style transferring Zhu et al. (2017), inpainting Demir & Unal (2018), image restoration and manipulation Pan et al. (2021). VAE architectures have also been successfully deployed in temporal prediction settings owing to their robustness properties Nazarovs et al. (2021); Rubanova et al. (2019).

Conceptually, in Score-based models, training data is transformed by slowly perturbing it using a Stochastic Differential Equation (SDE) with known steady state prior distribution. A parametrized model is learned to undo the perturbation process by solving the SDE backwards – so, the reverse-time SDE is approximated by a time-dependent neural network. Score-based models are now a popular choice amongst deep generative models, and have outperformed the previous state-of-the-art in image generation Dhariwal & Nichol (2021); Song et al. (2021); Ho et al. (2021). They have also been extensively used for video generation Ho et al. (2022), as well as Text-to-Image generation Nichol et al. (2021); Ramesh et al. (2022); Zhang et al. (2023).

**How to train Score-Based Models?** In score based models, the goal is to learn the *gradients* $\nabla_{\mathbf{x}} \log p_t(\mathbf{x})$ by training a time-dependent score based model $s_{\boldsymbol{\theta}}(\mathbf{x}, t)$. In particular we seek to solve the following optimization problem Song et al. (2021):

$$\boldsymbol{\theta}^* = \arg\min_{\boldsymbol{\theta}} \mathbb{E}_t \left[ \lambda(t) \mathbb{E}_{\mathbf{x}(0)} \mathbb{E}_{\mathbf{x}(t)|\mathbf{x}(0)} \left[ \left\| s_{\boldsymbol{\theta}}(\mathbf{x}(t), t) - \nabla_{\mathbf{x}(t)} \log p_t(\mathbf{x}(t) \mid \mathbf{x}(0)) \right\|_2^2 \right] \right], \tag{1}$$

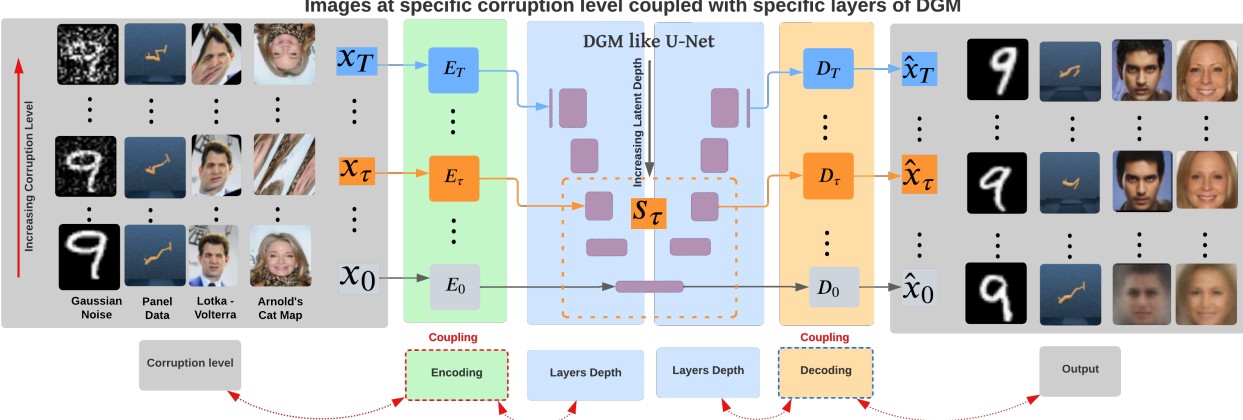

Figure 1: Coupling iterates from a given corruption process with Generators. The idea is to tie the encoder (E) and the decoder (D) of a Generator (G) to the level of noise present in the input.

where $p_t(\mathbf{x}(t) \mid \mathbf{x}(0))$ corresponds to the steady state prior density of the clean sample from training data $x(0)$, and $x(t)$ is the perturbed sample. We refer to the loss function in Equation (1) as the standard (or usual) loss function with $L(s_\theta, x(t), t)$ as the squared loss. For efficiently training Score-based DGMs, the prior distribution $p_t(\mathbf{x}(t) \mid \mathbf{x}(0))$ is assumed to be a Gaussian distribution since the mean and variance as a function of $t$ can be written in closed-form expressions that can be evaluated efficiently. We discuss more about the loss function and its connection to Kernel Density Estimation (KDE) in Appendix B.

**Problem Statement.** Non-Gaussian processes have many applications, especially in physics-informed neural networks Raissi et al. (2019); Pokkunuru et al. (2023). In this work we propose a novel way to train Score-based Generative models under non-Gaussian corruption procedures, thus incorporating physics based information directly during training and/or inference. We show how our novel technique Intermediate Generator Optimization can be used to increase the diversity of $G$ and solve various inverse problems when given the access to the forward process of SDEs where the forward operator cannot be computed easily.

**Our Contributions.** Motivated by the applications in various large scale vision settings we (i) propose a new fully differentiable framework that allows user specified corruption process within DGMs, and which can be trained using standard backpropagation procedures like SGD and its variants, (ii) analyze the statistical properties of our proposed framework called Intermediate Generator Optimization (IGO) using recently introduced Intermediate Layer Optimization (ILO) framework Daras et al. (2021) for solving inverse problems and finally, (iii) show the utility of our procedure with extensive results on several image generation and prediction tasks. In the challenging three dimensional point cloud setting, we identify potentially beneficial regularizers for improving the robustness profile of denoisers. We show how to apply our IGO framework to extend recently introduced projected power method (PPower) Liu et al. (2022) (code provided in supplement).

**Roadmap.** We start with a brief literature review in Sec. 2. In Sec. 3, we show how efficient discretization strategies can be used for non-Gaussian corruption processes and help improve range of generators. In Sec. 4, using the Information Bottleneck Principle, we introduce the notion of an intermediate generator regularizer for utilizing iterates of a discrete process during training. We argue that our regularizer can be efficiently optimized and analyze its sample complexity for linear inverse problems. Sec. 5 provides theoretical guarantees of the application of IGO. We then discuss various vision use cases of IGO such as image generation, dense extrapolation, and point cloud denoisers along with our experimental results in Sec. 6.

## 2 Literature Review

In recent years, there has been a growth in literature, which focuses on utilizing ODE and SDE theory in Neural Networks Chen et al. (2018); Li et al. (2020); Tzen & Raginsky (2019). More recently, physics-inspired approaches have been proposed to design algorithms for symbolic regression purposes that exploit the underlying symmetries in the problem. For example, Udrescu & Tegmark (2020) show that neural networks

can be used to learn reduce the search space by quickly finding such symmetries in the dataset. Similar ideas were also applied in the context of extending the range of generators for GAN/VAE with strategies that explicitly prevent mode collapse. Wasserstein based loss functions are often preferred if the DGM suffers from mode collapse during training Arjovsky et al. (2017). In fact, from a technical perspective, it is possible to show that Neural SDEs have the same expressive power as high dimensional GANs Kidger et al. (2021). The equivalence is achieved by simply parameterizing the (drift and diffusion functions of) forward process and its reverse process, using separate neural networks. This connection is mathematically interesting because it allows using such DGMs for solving classical statistical problems such as Monte Carlo simulations van Rhijn et al. (2021).

From a fixed generator, iterative refinement techniques were proposed for dense tasks, especially at high resolution Saharia et al. (2021). Score-based models generalize this idea, and achieve the state of the art results from many image synthesis problems Dhariwal & Nichol (2021). An interesting idea for Probabilistic Time Series Imputation was proposed by Tashiro et al. (2021). Authors explicitly train for imputation and can exploit correlations between observed values, unlike general score-based approaches. To extend the class of distributions that can be modeled within such a framework, various categorical and discrete parameterizations of the distributions have been proposed. For example, Austin et al. (2021) introduced Discrete Denoising Diffusion Probabilistic Models, which is based on corruption with transition matrices that mimic Gaussian kernels in continuous space, but based on nearest neighbors in embedding space, by utilizing absorbing states of a process.

On the VAE side, to deal with additional computational overhead of solving differential equations within score based models, authors in Gorbach et al. (2017) propose a scalable variational inference framework. Using coordinate descent on existing gradient matching approach, they propose new gradient matching algorithm that infers states and parameters in an alternating fashion, thus offering computational speedups in certain regimes. This is somewhat closer to our work than others since we infer the parameters of $E_\tau$ and $D_\tau$ during training which correspond to state dependent parameters in their setting. By relating ODEs with Gaussian Processes Wenk et al. (2019) provide a fast gradient matching procedure. On the other hand Bansal et al. (2022) discuss how the training and sampling procedures of diffusion models can be generalized for non-Gaussian corruption processes like blur and masking. We further develop on their ideas and explore the landscape of continuous non-Gaussian corruption processes, by using Lotka-Volterra Kelly (2016) and Arnold's Cat Map Bao & Yang (2012).

## 3 Non-Gaussian corruption via Discretization

Gaussian noise which are most popularly used in training diffusion models are very rarely corroborated in real-world noise and corruption patterns. The salt and pepper noises in images, artifacts in biological or medical images and the blurring/fogging of broadcast images are some examples of non-gaussian corruption. Non-affine processes also have many applications in physics based ML. Arnold Cat's map models rearrangement of configurations, so mass is conserved, whereas Lotka Volterra is nonlinear, so some mass escapes the (local) imaging based measurements of physical processes. Below we show how we use SDEs to model such non-affine corruptions processes.

**Basic Setup.** A Stochastic Differential Equation (SDE), where $f$ and $g$ are the drift and the diffusion functions respectively, and $w$ is the Standard Brownian Motion is represented by 2:

$$d\mathbf{x} = f(\mathbf{x}, t)\, dt + g(t)\, dw, \tag{2}$$

where $w$ is the Brownian Motion, and $dt$ is an infinitesimal negative timestep. We can map data to a noise distribution using the forward SDE in Equation (2) and reverse the SDE for generation using:

$$d\mathbf{x} = [f(\mathbf{x}, t) - g(t)^2 \nabla_\mathbf{x} \log p_t(\mathbf{x})]\, dt + g(t)\, dw. \tag{3}$$

After training using Equation (1), we may simply replace the log-likelihood in the reverse SDE by our model $s$ in Equation (3) for sampling purposes. In essence an approximate model $s$ will allow us to generate diverse, yet realistic samples using a small amount of independent random noise component, $g$. Later in this section, we present a discussion about why the SDE-based loss function automatically induces diversity in samples generated by a generative model.

**Our Assumption on Forward Process.** For the SDE model given in Equation (2), typically the drift function $f$ is chosen to be affine, thus bypassing the need to obtain $p_t(x(t) \mid x(0))$. We consider learning the trajectory information under cases when $f$ is not necessarily affine. We do this by modelling $f$ using two popularly known processes, Lotka-Volterra and Arnold's Cat Map (samples shown in Figure 1, more in supplement). In order to do that, we relax the assumption that we have access to an efficient discretization of the process in Equation (2). Specifically, we say that the forward SDE Equation (2) can be efficiently computed using the Euler-Maruyama (EM) method. We will use $\tilde{\mathbf{x}}$ to denote the approximation to $\mathbf{x}$ provided by the EM discretization. For a time-varying multiplicative process defined by $d\mathbf{x}_t = a(\mathbf{x}_t, t) \, dt + b(\mathbf{x}_t, t) \, dw_t$, where $a$ and $b$ are smooth functions similar to $f$ and $g$ in Equation (2), but $a$ is not affine, the EM algorithm executes the following iterations:

$$\tilde{\mathbf{x}}_{t+\Delta t} = \tilde{\mathbf{x}}_t + a(\tilde{\mathbf{x}}_t, t)\Delta t + b(\tilde{\mathbf{x}}_t, t)z_t\sqrt{\Delta t}, \tag{4}$$

where $\tilde{\mathbf{x}}_t$ denotes the approximation at time t, and $z_t \sim \mathcal{N}(\mathbf{0}, \mathbf{I})$ is a sample drawn uniformly at random from the standard normal distribution. Note that the process defined in Equation (2) satisfies our assumption here since we allow $b$ to depend on $x_t$. The discretization provided by Equation (4) enables us to simulate the SDE and subsequently sample from $p_t(x(t) \mid x(0))$.

**SDEs for Modeling Diversity.** In order to improve the range of generators as desired, strategies that involve explicit modeling of the *forward* or the corruption process have been suggested. Mathematically speaking, this can be done by using Stochastic Differential Equations (SDE) of the form shown in Equation (2).

For generation, $\mathbf{x}(0)$ represents the (clean) dataset to be generated, given samples $\mathbf{x}(T)$ for some sufficiently large $T$. We know that under standard assumptions on $f$, we have that: as $T \uparrow \infty$, $\mathbf{x}(T)$ mostly represents noise, that is, has least information. Correspondingly, given samples $\mathbf{x}(0)$, the learning problem is to train a generator $G$ so that it produces a randomized version of $x(0)$ from random noise $\mathbf{x}(\infty)$. The key property of SDEs that is attractive for distribution $G$ is that each equilibrium solution of a forward SDE will have its own *unique* trajectory, which can be used during training. We do so by coupling the intermediate iterate $x_\tau$ during training with specific layers in a DGM, as shown in Figure 1.

## 4 Efficient DGMs for Discrete Processes

To provide guarantees on DGM-based downstream tasks such as solving inverse problems, we propose a new loss function to train the score-based model "$s$" based on individual trajectories of samples corrupted by the discretized process defined in Equation (4). We begin by writing the empirical finite sample form of the optimization problem in Equation (1) as follows:

$$\min_{\boldsymbol{\theta}} \sum_{\tau=t_1}^{t_T} \lambda(\tau) \sum_{i=1}^{N} \mathbb{E}_{\mathbf{x}_i(\tau)|\mathbf{x}_i} L\left(s_\theta, \mathbf{x}_i(\tau), \tau\right), \tag{5}$$

where $T$ represents the discretization size, and $0 \le t_i \le 1 \ \forall \ i = 1, \dots, N$. To solve the optimization problem in Equation (5), the most popular choice is to use first-order backpropagation type methods. Importantly, the worst-case complexity of solving Equation (5) scales linearly with the discretization size $T$, which is intractable in large-scale settings. By the chain rule, the efficiency of such algorithms strongly depends on the ease of evaluating the derivative of the forward operator, denoted by the conditional expectation. While previous works assume that there is a closed-form solution to evaluate the conditional expectation, such an assumption is invalid in our examples discussed above. If $T = 1$, then we may simply simulate the dataset $\{\mathbf{x}_i\}$ using Equation (4) to obtain $\tilde{\mathbf{x}}_i(t_1)$, and use it to compute the loss, and backpropagate.

**Handling $T = 2$ case using IBP.** Now we consider the setting in which we have access to only one intermediate iterate $\tilde{\mathbf{x}}_\tau$ where $\tau < t$ for some arbitrary $t \sim \mathcal{U}(0, 1)$, in our trajectory. Naively, we can train two DGMs in parallel, one each for $\tilde{\mathbf{x}}_t$ and $\tilde{\mathbf{x}}_\tau$, thus incurring twice the memory and time complexity, including resources spent for hyperparameter tuning. We propose a simpler alternative through the lens of the so-called Information Bottleneck Principle (IBP) – the de-facto design principle used in constructing standard Autoencoders architectures such as U-Net. In feedforward learning, the main result due to IBP is that layers

in a neural network try to *compress* the input while maximally preserving the relevant *information* regarding the task at hand Goldfeld & Polyanskiy (2020). For designing neural networks, this corresponds to choosing a sequence of transformations $T_l$ such that the distance between successive transformations $d(T_l, T_{l-1})$ is not too big. In practice, we can ensure this by simply choosing dimensions of layerwise weight matrices by a decreasing function of layers (or depth). Intuitively, the idea is that if dimensions of $T_l$ are a constant factor of $T_{l-1}$, then at optimality (after training), when random input passes via $T_l$, a constant factor of noise available when it passed through $T_{l-1}$ will be removed, in expectation.

By using the correspondence between the forward process defined in Equation (4) and intermediate layers in a DGM, we will now define our loss function for intermediate iterates $\tilde{\mathbf{x}}_\tau$. For simplicity, we drop the subscript **i**. We define the regularization function $\mathcal{R}$ as follows,

$$\mathcal{R}(\tilde{\mathbf{x}}_\tau, t) = L\left(D_\tau \circ s_\tau \circ E_\tau, \tilde{\mathbf{x}}_\tau, t\right), \tag{6}$$

where $s_\tau$ (see dark green box in Figure 1) denotes the *restriction* of the score-based model $s$ to iterate $\tilde{\mathbf{x}}_\tau$, $E_\tau$ denotes a shallow encoder for $\tilde{\mathbf{x}}_\tau$, and similarly for the decoder $D_\tau$. We will refer to the DGM defined by $D_\tau \circ s_\tau \circ E_\tau$ as the **Intermediate Generator**, see Figure 1. Intuitively, the regularization function $\mathcal{R}$ is defined to modify the parameters of the model $s$ to a specific set of connected layers given by $s_\tau$. To see this, note that the overall parameter space $\boldsymbol{\theta}$ can be seen as a product space over layers $\theta_l$. At any given training iteration, the $\mathcal{R}$ function *restricts* the update to layers that are suitable for decoding $\tilde{\mathbf{x}}_\tau$. Thus, for $t \sim [0, T], \tau < T$, we can use Equation (6) to rewrite the training objective from Equation (1) as:

$$\arg \min_{\theta, E_\tau, D_\tau} \mathbb{E}_t \left[ \lambda(t) \mathbb{E}_{x(0)} \mathbb{E}_{(x(t)|x(0))} L(s_\theta, x(t), t) + \lambda(\tau) \mathbb{E}_{x(0)} \mathbb{E}_{(x(\tau)|x(0))} L(D_\tau \circ s_{\tau,\theta} \circ E_\tau, x(\tau), t) \right] \tag{7}$$

This new training objective allows us to use a convex combination of scores of standard decoder $D_T$, and intermediate iterate decoder $D_\tau$. All our experiments in Sec. 6 utilize this convex combination objective function for training. We can also extend Equation (7) to multiple iterates ($T > 2$). When we are given more than one intermediate iterate from the discretized EM algorithm, Equation (6) can be rewritten as:

$$R_0^T(\tilde{\mathbf{x}}) = \sum_{\tau=t_1}^{t_T} L\left(D_\tau \circ s_\tau \circ E_\tau, \tilde{\mathbf{x}}_\tau, T\right). \tag{8}$$

**Interpreting $\mathcal{R}$.** Using a small step size in EM guarantees us that a larger class of SDEs can be simulated (see Neuenkirch et al. (2019)), so $\tilde{x} \sim x$. Fix a sufficiently small $\Delta t$, so $\tilde{\mathbf{x}} \approx \mathbf{x}$ almost everywhere at any time $t$. When the forward corruption process is a *Markov* process, increments of $g$ are independent of $t$, and the intermediate iterates do not carry additional information. However, in our discrete setting, the drift $a$ and diffusion $b$ functions are time-varying for Arnold's Cat Map and Lotka-Volterra. In this case, $\mathcal{R}$ simply tries to revert the time-dependent noise process in the relevant part of the network $s_\tau$ using appropriate $E_\tau$, and $D_\tau$. Our regularization function provides time-dependent noise information explicitly by using the intermediate iterate $\tilde{\mathbf{x}}_\tau$ during training in appropriate parts of the DGM. This ensures that our IGO framework can be extended to energy-based models, which are known to perform as well as Score based DGMs Salimans & Ho (2021) for image generation.

**Implementation.** IGO utilizes fewer computational resources in the following sense: if $E_\tau$, and $D_\tau$ are relatively shallow networks (as shown in Figure 1), then the cost of computing gradients with $\tilde{\mathbf{x}}_\tau$ is negligible compared to $\tilde{\mathbf{x}}$, thus achieving cost savings by design. Using intermediate iterates while training $s_\theta$ provides more information of the corruption process by mapping multiple points to the same trajectory. Another clear benefit that IGO has over methods like Sliced Score Matching Song et al. (2019) while learning the scores of a non-affine drift is the easy sampling approach. Using IGO provides two different gradients viz., $G_T$, and $G_\tau$, thus expanding the range of the reverse SDE. In Sec. 6, we show how the two different scores learned by our model can be utilized to generate different samples by a simple modification over existing sampling methods.

## 5 Theoretical Analysis for Downstream Tasks

We utilize the key result in ILO to analyze our procedure for downstream tasks. To that end, we assume that the parameters of $s$, $E_\tau$ and $D_\tau$ are fixed (given by a pre-trained model), and show that IGO is suitable for

solving inverse problems under low sample settings. The input of the ILO algorithm is pre-trained DGM with the goal of tuning the noise distribution for better image generation using gradient descent schemes. In IGO, we use the knowledge of the forward process to optimize the parameters of the DGM to increase its range **during** training.

**Necessary Condition on DGM.** We follow the same observation model as in Daras et al. (2021) given by $y = Ax + \mathfrak{n}$ where $\mathfrak{n}$ is a random variable representing noise, and $x$ is the unknown. It is well known that when the measurement matrix $A$ satisfies certain probabilistic requirements, then it is possible to solve for $x$ using a single (sub)gradient descent scheme Candes & Tao (2006); Tropp (2015). This is specified using the S-REC condition in DGM-based prior modeling using the CSGM algorithm Bora et al. (2017). An example that satisfies the probabilistic requirements is when the entries in $A$ are distributed according to Gaussian distribution, as mentioned in Daras et al. (2021).

To analyze $\mathcal{R}$ for prior modeling purposes, we assume that after training the generator has a compositional structure given by $G_1 \circ G_2$, which is followed in most standard architectures. The following observation is crucial in analyzing IGO for compressive sensing purposes.

**Observation 1. (Range expansion due to $E_\tau$.)** *Setting the intermediate generator (of s) to be $G_\tau$, the range of our pre-trained model, $G_1$ is increased to,*

$$\underline{G} := G_\tau(B_2^k(r_1)) \oplus E_\tau(B_2^{k_\tau}(r_\tau)), \tag{9}$$

*where $\oplus$ denotes the Minkowski Set sum, and $k_\tau$ corresponds to the dimension of intermediate code associated with $\tilde{\mathbf{x}}_\tau$.*

With the increased range in Observation 1, we show that using the overall generator trained with IGO as a prior can be sample efficient in the following lemma. We make the same assumptions as in Theorem 1. in Daras et al. (2021) on the entries of $A$, and smoothness of $G_2$.

**Lemma 1.** *Assume that the Lipschitz constant $\underline{G}$ is $\underline{L}$, and we run gradient descent to solve for the intermediate vector, as in CSGM. Assume that the number of measurements $m$ in $y$ is at least*

$$\min(k \log(L_1 L_2 r_1/\delta), k_\tau \log(\underline{L} L_2 r_\tau/\delta)) + \log p, \tag{10}$$

*where $p$ is the input dimension of $G_2$ or the intermediate vector. Then, for a fixed $p$, we are guaranteed to recover an approximate $x$ with high probability (exponential in $m$).*

Please see the supplement for the proof of the signal recovery lemma 1. Our proof uses the (now) standard $\epsilon$−net argument over the increased range space Vershynin (2018). In essence, if the Lipschitz constant of the intermediate generator $\underline{L}$ is lower than the Lipschitz constant of $G_1$, then we get an improved sample complexity bound for IGO. Intuitively, lemma 1 states that the number of samples required during downstream processing depends on two factors. The *latent dimension size* denoted by the $\log p$ term which remains the same with or without intermediate iterates, and *smoothness* denoted by the Lipschitz constants $L_1, \underline{L}, L_2$.

While we have no control over the size $p$, if the intermediate generator is smoother, and performs well on training data $y$, then gradient descent succeeds in recovering the "missing" entries with fewer samples. In the case when intermediate iterates are not useful, then our framework allows us to simply ignore the intermediate generator without losing performance. In other words, our framework preserves the hardness of recovery – easy problems remain easy.

### 5.1 Improving Generative PCA.

To utilize Lemma 1 to the problem of PCA with complex generative priors, we modify the variational formulation of PCA to,

$$\mathbf{v} := \arg \max_{\mathbf{Z} \in \mathbb{R}^n} \mathbf{Z}^T V \mathbf{Z} \quad \text{s.t.} \quad \mathbf{Z} \in \text{Range}(G_T \cup G_\tau). \tag{11}$$

In Equation (11), input $V$ is the covariance matrix of the dataset, and the output $\mathbf{v}$ is the principal component. We may initialize $Z \in \mathbb{R}^n$ by randomly sampling or by choosing the column corresponding to the largest

---

**Algorithm 1** Solving PCA using IGO's generative priors

---

**Input** - Generators $G_T, G_\tau$, Covariance matrix $V$

1: **for** $i \in \{\tau, T\}$ **do**
2:  Initialize $\mathbf{Z}_i^{(0)}$ (randomly or manually)
3:  **repeat**
4:   Compute $\mathbf{v}_i^{(t)} = \mathbf{Z}_i^{(t)T}\mathbf{V}\mathbf{Z}_i^{(t)}$
5:   Update $\mathbf{Z}_i^{(t+1)}$ by solving: $\max_{\mathbf{Z} \in R^n} \mathbf{Z}_i^T \mathbf{V}\mathbf{Z}_i$ subject to $\mathbf{Z}_i \in \text{Range}(G_i)$
6:  **until** Convergence
7: **end for**

**Output** - $\mathbf{v}_i^{(t)}$, the reconstructed top principal eigenvectors of $V$

---

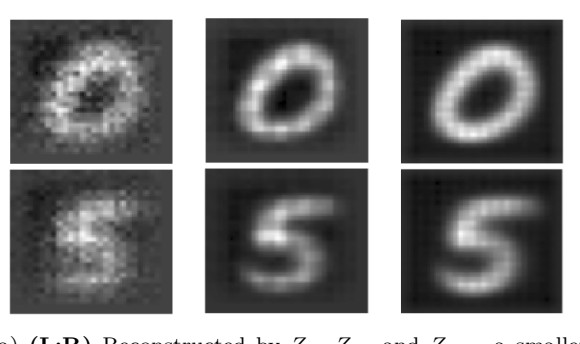 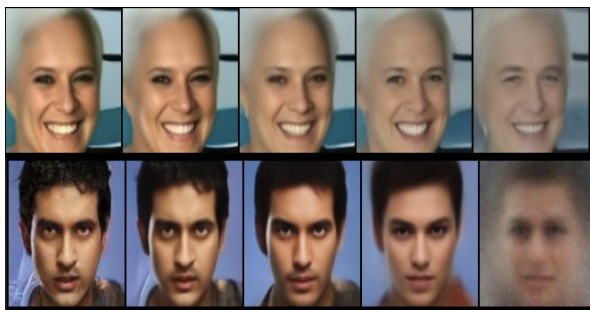

(a) **(L:R)** Reconstructed by $Z_\tau$, $Z_T$, and $Z_{\tau'}$ – a smaller dimensional version of $Z_\tau$ respectively.

(b) Generated Samples, using $\alpha = 0.8$ and Lotka-Volterra as the non-affine drift, in Sec. 6.2

Figure 2: Image generation using IGO, Sec. 6.1 and Sec. 6.2

diagonal entry of $V$. In our implementation, we compute the eigenvectors using the generators $G_T$ and $G_\tau$ in decomposed manner.

**Leveraging a diverse $G$ for generic Downstream Tasks.**  Consider a feature extractor (or an encoder) induced by an appropriate SDE-guided DGM, that can extract *relevant* features from a mixture of two distributions, for example, training, and adversarial. In this setting, if we know the distribution from which a given feature has been sampled, then, we can hope to predict the sample at optimal error rates by using a neural network with sufficient layers. When the mixtures have a natural dependency (given by SDE in Equation (2)), then we may simply use weight sharing instead of training two separate models. Assume that a DGM has a total of $P$ parameters. Then, training $|T|$ DGMs each for a time step of a discretization size $|T|$ would require $O(P|T|)$ parameters. However in our approach, say we require $p$ extra parameters for $E_\tau$, $D_\tau$, then the total number of parameters in our IGO framework is $O(P + (p-1)|T|) = O(P + |T|)$, for small values of $p$ – as is the case in our experimental settings – this can be a huge reduction in the number of parameters $O(P|T|)$ in standard framework vs $O(P + |T|)$ in our IGO framework. Thus, achieving memory as well as some time savings. We touch upon this in Sec. 6.4.

## 6 Experiments

In this section, we explore different applications of IGO across varying setups to showcase our idea of utilizing an intermediate iterate $\tilde{\mathbf{x}}_\tau$ ($\tau = t/2$ in all cases). First, in Sec. 6.1 we show our IGO framework for Generative PCA, using a pre-trained network. Next, we show the procedure to train IGO in a U-Net based architecture in Sec. 6.2. In Sec. 6.3, we tackle the challenging trajectory prediction tasks, in particular, we modify the ODE to take inputs from the intermediate, and final iterates during training. Finally, In Sec. 6.4, we demonstrate the use of IGO in a *dense* prediction task of denoising posed as a supervised learning problem. Here, a multi-layer perceptron is utilized as a score-based model for denoising purposes.

### 6.1 Generative PCA with IGO

#### 6.1.1 Setup Details

Here, we use a pre-trained network capable of generating MNIST digits to solve the eigenvalue problem using the Projected Power (PPower) method Liu et al. (2022). The basic idea of the classic power method for the eigenvalue problem is to repeatedly apply the eigenvector matrix to a starting vector (noise) and then normalize it to converge to the eigenvector associated with the largest eigenvalue. The key difference between the PPower method, compared to the classic power method is that it uses an additional projection operation to ensure that the output of each iteration lies in the range of a generative model. We perform experiments by optimizing the noise vector in 3 different cases, using Algorithm 1. Firstly, we initialize the noise vector, Z, and optimize it using the intermediate generator for 2 cases. We term these instances of Z, $Z_{\tau\prime}$ and $Z_\tau$. For $Z_\tau$, we use $D_\tau$, the last decoding layer of the intermediate generator. Whereas for $Z_{\tau\prime}$, the decoder of $S_\tau$, as shown in Figure 1, which is at a smaller dimension,is used to constrain the PCA search space. Finally, we optimize the noise vector $Z_T$, for generation at $D_T$, the last decoding layer of the usual generator. In this experiment we show the utility of optimizing the noise vector in smaller dimension ($Z_{\tau\prime}$) by comparing it to the noise vector for the usual generator ($Z_T$) and the intermediate generator ($Z_\tau$). The projection step for each case is solved by using the generated samples from the respective decoders using an Adam optimizer with 100 steps and a learning rate of 0.001.

#### 6.1.2 Experimental results

Firstly, we see from Figure 2 - (a), that our model was able to generate images in all the 3 cases. The image generated by reconstructing $Z_{\tau\prime}$, the leftmost in Figure 2 - (a), looks noisier since the model's features at that stage were smaller in comparison to the other cases. We also analyze the cumulative explained variance as a function of the number of components and show the results in the supplement. We see that the images generated using a smaller dimension of noise vector $Z_{\tau\prime}$ use less principal components to explain variance compared to $Z_T$ and $Z_\tau$.

**Takeaway.** By Theorem 3 in Liu et al. (2022) we know that the reconstruction error is proportional to the square root of the dimensions of Z. Our experiments indicate that the smaller dimensions of Z in $Z_\tau$ and $Z_{\tau\prime}$ can also be used to generate images. Thus, giving us lower reconstruction error as well as parameter savings of approximately $Z_T$ - $Z_{\tau\prime}$ dimensions.

### 6.2 IGO for training DGMs with non-Gaussian Drift

#### 6.2.1 Setup Details

In a standard score-based generation, the transition kernel is assumed to be a multivariate Gaussian distribution. Given a clean image $\mathbf{x}(0)$, we can apply the kernel using closed form evaluation to obtain the noisy image $\mathbf{x}(t)$ for random $t \sim \mathcal{U}(0,1)$. Neural networks are then used to estimate the score, also known as the time-dependent gradient field, to reverse the corruption process.

Here, we start off with explicitly modeling our corruption process using a non-affine SDE with Arnold's cat map and Lotka-Volterra as the drift to attain an intermediate iterate $\tilde{\mathbf{x}}_\tau$, by setting up Eqn. (4). Our goal here is to utilize our proposed Intermediate Generator Optimization, for training purposes. To do so, we simulate the forward SDE till some random time $t$ and store an (additional) intermediate iterate for every trajectory to obtain $\mathbf{x}_t$ and $\tilde{\mathbf{x}}_\tau$ ($\tau = t/2$). The two iterates are then used to set the training objective based on Eqn. (5) with our proposed regularizer in Eqn. (6). As per Figure 1, the intermediate iterate $\tilde{\mathbf{x}}_\tau$ has its own intermediate pathway $s_\tau$ into our overall model $s$.

#### 6.2.2 Experimental results

We choose the CelebA dataset to test the generative capabilities of our IGO formulation. Similar to Song et al. (2021) we also use a U-net Ronneberger et al. (2015) architecture as the backbone of our model. The parameters of which are shared across time using sinusoidal position embeddings Vaswani et al. (2017).

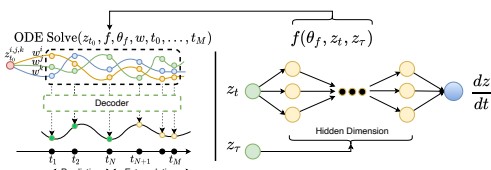

(a) Incorporation of intermediate iterates in Mixed Effect Neural ODE for analyzing the dynamics of panel data.

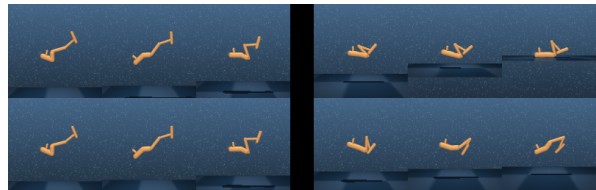

(b) One of the Hopper samples: **Top** is the true data, **Bottom** is the prediction. Before the black line is observed data; after it are the extrapolated samples previously not seen by the model.

Figure 3: IGO on Panel Data, Sec. 6.3

During the forward pass, our simulated iterates $\mathbf{x}_t$ and $\tilde{\mathbf{x}}_\tau$ are passed into the network, using their specific pathways, refer Figure 1. For the iterate $\tilde{\mathbf{x}}_\tau$ we use the intermediate encoder $E_\tau$ and an intermediate decoder $D_\tau$. As discussed in Sec. 5.1, since we train the encoders for both the iterates $\mathbf{x}_t$ and $\tilde{\mathbf{x}}_\tau$ together, we save the number of extra parameters that a model would have needed to specifically train each iterate.

The iterates $\mathbf{x}_t$ and $\tilde{\mathbf{x}}_\tau$ are jointly used to train the network using a convex combination of loss. We use the scalar $1 - \alpha$ to refer to the weight given to the gradient of the regularization function $\mathcal{R}$, Equation (6). Thus, the lesser the value of $\alpha$, the more the weight of the intermediate iterates in the objective function for training the score network. The trained score model is then provided to the RK45 ODE Solver (*scipy*) for sampling. The top row of Figure 2 - (b) shows the progressive samples generated using the intermediate layers, while the bottom row shows the samples from the usual layers. The rightmost images in Figure 2 - (b) are the closest to noise. The model used to generate images in Figure 2 - (b) was trained using $\alpha = 0.5$. We utilize the PyTorch implementation of Fretchet Inception Distance (FID) Heusel et al. (2017) score provided by Seitzer (2020). The images from the intermediate layers have an average FID of 3.5, while the images from the usual layers have an FID of 3.3, similar to Song et al. (2021). All the comparisons were made using 192 feature dimension and the model with 4 residual blocks per resolution. We provide results on the effect of $\alpha$, using results on MNIST data in the supplement.

**Takeaway.** Our results show the advantage of using the IGO framework, as the intermediate layers are able to generate images of almost similar quality (FID) as the usual layers, while providing a shorter pathway for lesser corrupted iterates of a noise process.

### 6.3 IGO on Panel Data

### 6.3.1 Setup Details

One of the applications of our proposed Intermediate Generator Optimization is modeling the trajectory of a Differential Equation, which describes a dynamic process. We use the setup in Nazarovs et al. (2021). That is, we consider that the dynamic process is modeled by changes in the latent space $z$ as $\dot{z}(t) = D(z, t)\mathbf{w}$, where $D(z, t)$ is a neural network and $\mathbf{w}$ is a corresponding mixed effect (a random projection describing flexibility (stochasticity) of dynamics). We propose to extend $D(z, t)$ with our IGO, as $D(z_t, z_\tau)$ using a neural network $f$, see Figure 3 - (a) for details on loss computation with $z_\tau$.

### 6.3.2 Experimental results

We evaluate our approach on two temporal vision datasets, representing the dynamics of panel data. Namely, we apply our proposed method to the variations of MuJoCo Hopper and Rotation MNIST. As baseline methods, we choose ODE2VAE for MuJoCo Hopper, and NODE & MEODE for Rotation MNIST, as per Nazarovs et al. (2021). All the baselines use an ODE backbone to analyze panel data, but with a different approach of modeling the uncertainty. We evaluate our model against those baselines on interpolation and extrapolation, and use MSE for comparison.

**MuJoCo Hopper** The dataset represents the process with simple Newtonian physics, which is defined by the initial position, velocity, and the number of steps. To generate data we randomly choose an initial

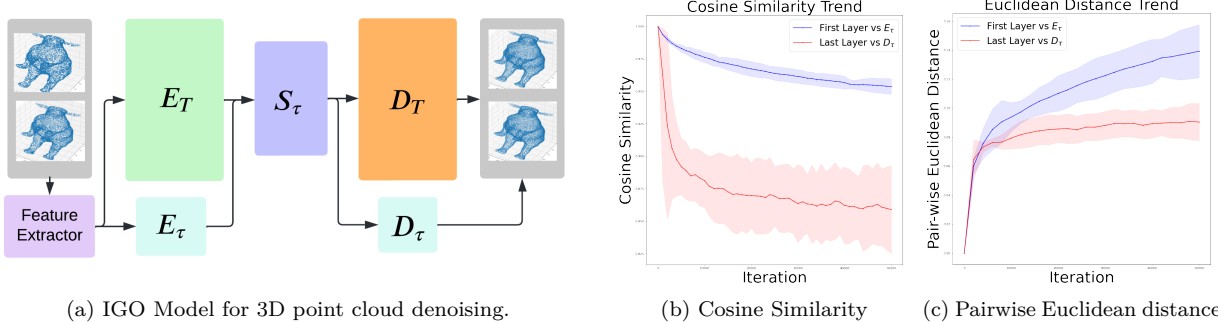

(a) IGO Model for 3D point cloud denoising.    (b) Cosine Similarity    (c) Pairwise Euclidean distance

Figure 4: IGO in 3D Processing. Figure **(a)** shows the utilized architecture.; Figures **(b)** and **(c)** compare the weights of $E_T$ against $E_\tau$, and $D_T$ against $D_\tau$.

position and a sample velocity vector uniformly from $[-2, 2]$. We evaluate our model on interpolation (3 steps) and extrapolation (3 steps) and provide a visualization of one of the experiments in Figure 3 - (b). The MSE for interpolation was 0.0289, while for Extrapolation it was 0.2801. In comparison ODE2VAE Nazarovs et al. (2021)'s interpolation MSE is at 0.0648.

**Rotation MNIST**   We evaluate our approach on a more complicated version of the Rotating MNIST dataset. We construct the dataset by rotating the images of different handwritten digits and reconstructing the trajectory of the rotation in interpolation and extrapolation setups. For a sampled digit we randomly choose an angle from the range $[-\pi/4, \pi/4]$ and apply it at all time steps. In addition, to make our evaluation more robust and closer to a practical scenario, we spread out the initial points of the digits, by randomly rotating a digit by angles from $-\pi/2$ to $\pi/2$. The generated 10K samples of different rotating digits for 20 time steps were split into two equal sets: interpolation and extrapolation. Like the previous experiment, MuJoCo Hopper, we evaluate our model on interpolation (10 steps) and extrapolation (10 steps). In this case, the MSE for interpolation and extrapolation were 0.0082 and 0.1545 respectively. In comparison NODE Nazarovs et al. (2021), the MSE is 0.0074 and 0.1661 respectively. Whereas MEODE Nazarovs et al. (2021), the MSE is 0.0057 and 0.1641 respectively. Refer to the supplement for implementation details.

**Takeaway.**   Clearly, by introducing IGO in the model, we achieve good generation ability even for extrapolation in the future time steps.

### 6.4   IGO in 3D Processing

### 6.4.1   Setup Details

We use a standard deep learning-based point cloud denoiser that comprises a Feature Extraction Unit and a Score Estimation Unit. The feature extractor is tasked to learn local as well as non-local features for each point in the noisy point cloud data, provided as the input. The score estimator provides point-wise scores, using which the gradient of the log-probability function can be computed. We adopt the same feature extraction unit as well as the Score Estimation Unit used in Luo & Hu (2021). Here we introduce intermediate iterates to test if we can generate different versions of the denoised samples using a pretrained score-based model provided by Luo & Hu (2021), see Figure 4 - a).

### 6.4.2   Experimental results

At the beginning of the denoising step, we have the given noisy input, $\mathbf{x}_t$ and its intermediate iterate $\tilde{\mathbf{x}}_\tau$. Our architecture is modelled using a 3-layer MLP as in Luo & Hu (2021), and the newly added intermediate layers, $E_\tau$ and $D_\tau$. The newly added layers replicate the already present first and last layer perceptrons but are only dedicated to the intermediate layers. We initialize the weights of $E_\tau$ and $D_\tau$ to be half of the weights of the pre-trained layers. The training was done using a convex combination of loss, defined using the intermediate and final iterate, $\mathbf{x}_t$ and $\tilde{\mathbf{x}}_\tau$. The architecture can be seen in Figure 4 - (a). We use $\alpha$ to denote this convex combination hyperparameter. The smaller the value of $\alpha$, the more the weight of the intermediate

iterates in the objective function for training the score network. The network was tested using the PU-Net test set provided by Luo & Hu (2021). We provide comparisons to different baselines using Chamfer Distance (CD) Fan et al. (2016) and Point-to-Mesh Distance (P2M) Ravi et al. (2020) as metrics in the supplement.

### 6.4.3 Finding new Generators using $X_\tau$

Here we validate our hypothesis that we can train diverse generators using the intermediate iterate. In order to quantify the spread of our overall model, we use the cosine similarity metric suggested for generalization purposes in Jin et al. (2020). Here, we compute the cosine similarities between the weights of $E_T$ and $E_\tau$ and between $D_T$ and $D_\tau$. Recall, $D_T$ is the usual decoder, which is the last layer, and $D_\tau$ is its corresponding intermediate decoder. We also calculate the pair-wise Euclidean distance between the respective weights; see Figure 4 - (b), (c).

**Takeaway.** We observed that as we fine-tune our model the cosine similarity between $D_T$ and $D_\tau$ decreases by 10% and the Euclidean between $E_T$ and $E_\tau$ increases by 15%, in $50k$ iterations —two dissimilar decoders that perform well on the training dataset.

## 7 Conclusions

From a theoretical perspective, our results indicate that it is indeed possible to be more sample efficient while solving inverse problems using our IGO construction by slightly modifying the standard assumptions made on DGMs. We can implement our regularizer on any end-to-end differentiable DGM with minor code modifications as shown in two 2D images and one 3D point cloud setting. Our experiments, in the case of eigenvalue problems also show that our framework and the landscape of its parameters has interesting properties and can be exploited for efficient optimization purposes and parameter savings. We show how intermediate iterates, which are implicitly generated when using iterative algorithms or multi-step noise conditioning processes, better aid DGMs by providing additional structure. We believe that abstract ideas from dynamical systems are very much relevant in decision-making in vision settings as more and more pretrained models are deployed on form factor and/or edge devices to make on-the-fly decisions. Our results provide us with a shred of affirmative evidence that ideas from dynamical systems will be of utmost importance when each such decision has nontrivial consequences (say due to the presence of adversarial noise).

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

## Appendix

We include several appendices with proofs, additional details and results of our experiments. First, in Appendix A we show the proof for Lemma 1. We then provide the implementation details for using our proposed non-Gaussian drift functions in Appendix C. In Appendix D, we show additional results, along with pseudocode for the experiments in Sec. 6. We also provide results from an additional experiment on training DGMs with IGO on the MNIST dataset in Appendix D.2.2.

## A   Proof of signal recovery Lemma 1

*Proof.* Our proof follows the strategy detailed in Daras et al. (2021). In particular, we will use the metric entropy concentration inequality for $\ell_2$ balls, and Maurey's empirical method (the Sudokov minorization inequality) for $\ell_1$ ball, which are more accurate than the standard $\epsilon$-net argument. We now explain the details.

In order to make any sample complexity guarantee using a pretrained generator, we have to assume that we have access to an algorithm that can solve inverse problem using a pretrained generator. For this purpose, we will use the CSGM method for solving inverse problems using the generators that we have trained (that includes the intermediate iterates). Observe that the intermediate generator has its own noise source that can be optimized using the CSGM method for solving inverse problems. Hence, after training, we obtain two separate generators, i.e., two independent noise sources. Following the notations in the main paper, for any decomposition of the overall $G = G_1 \circ G_2$, we can define the respective true optimum in the extended range of the intermediate generator $G_\tau := s_\tau \circ D_\tau$ be given as,

$$\bar{z}_\tau^p = \arg \min_{z^p \in G_\tau(B_2^k(r_1)) \oplus B_1^p(r_2)} \|x - G_2(z_p)\|, \tag{12}$$

and its corresponding measurements optimum in the extended range of $G_\tau$ be given by $\tilde{z}_\tau^p$. We will drop the superscript $p$ to avoid notation clutter. Then our results follow by noting that this intermediate generator $G_\tau$ satisfies the S-REC condition since the corresponding intermediate encoder $E_\tau$ and decoder $D_\tau$ are Lipschitz. That is, following inequality (78) in Daras et al. (2021), we have that,

$$\|G_2(\bar{z}_\tau) - G_2(\tilde{z})\| \leq \frac{4\|G_2(\bar{z}) - x\| + \delta_\tau}{\gamma}, \tag{13}$$

where $\delta_\tau$ is a constant that depends (at most polynomially) only on latent dimensions and $p$. We can now take the minimum of the sample complexities of both generators since they each have independent noise sources. Finally, we have the desired result as claimed due to the S-REC property for the nested $\ell_1$ ball of the intermediate generator. $\qquad\square$

## B   Connection to KDE.

To understand the loss function in Equation (1) consider the standard Kernel Density Estimation (KDE) procedure for the moment. KDE is a smooth estimator of probability density function, and is known to converge faster than histograms (function values), in terms of sample complexity, see Wasserman (2010). While parameters of KDEs are usually tuned to maximize the likelihood of observed data, using higher order (or finer) information such as gradients have been studied recently in small scale settings, see Sasaki et al. (2017); Kim et al. (2019). Note that if $x(t) \in \mathbb{R}^n$, then $\nabla_{x(t)} \log p_t(x(t) \mid x(0)) \in \mathbb{R}^n$, whereas $\log p_t(x(t) \mid x(0)) \in \mathbb{R}$, so training $s_\theta$ to fit gradients $\nabla_{x(t)} \log p_t(x(t) \mid x(0))$ may be slightly more challenging than simply maximizing the likelihood $\log p_t(x(t) \mid x(0))$.

In Score based DGMs, the generator $G$ learns the *gradient* of the log-likelihood function explicitly instead of the density function itself. There are two benefits associated with this approach: (i) statistical range of the generator can be naturally improved during sampling since gradients are often more informative (smooth function values can be approximated using gradients by Taylor's theorem), and (ii) computationally tractable alternative to kernel density estimation since it does not require us to estimate the normalizing constant which is usually intractable in high dimensions.

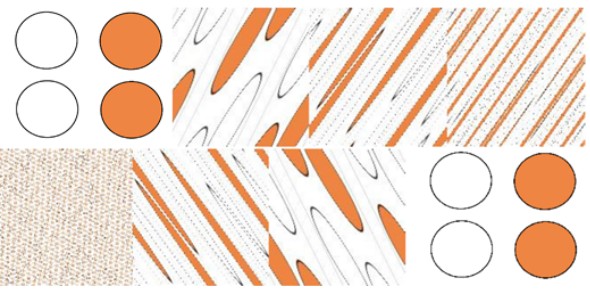
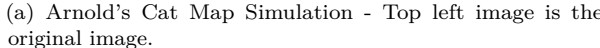
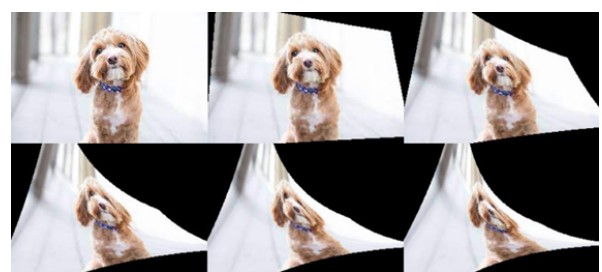

(a) Arnold's Cat Map Simulation - Top left image is the original image.

(b) Lotka-Volterra Simulation - Top left image is the original image.

Figure 5: Data augmentation schemes.

## C   Euler–Maruyama Algorithm to aid Data Augmentation

Here we show the application of EM Algorithm on two simulatable processes, Lotka-Volterra and Arnold's Cat Map. We use scipy.integrate.solve_ivp Virtanen et al. (2020) to integrate our differential equations and get the trajectory for each pixel in the image, thereby finding the trajectory of the image as a whole. The resulting images are then plotted using Matplotlib Hunter (2007). An image with 2 colored balls and 2 non-colored balls following the trajectory of Arnold's Cat Map is shown in Figure 5 - a.). Figure 5 - b) shows the simulation of a Dog following the trajectory of Lotka-Volterra. The code to reproduce the images in Figure 5 can be found in the provided zipped folder.

## D   Experiments

In this section we showcase additional results from the experiments performed to explore the applications of IGO.

### D.1   Generative PCA with IGO

#### D.1.1   Implementation Details

We show the reconstructed images for digits 0, 1, 5 and 9 from $Z_T$, $Z_\tau$ and $Z_{\tau'}$ in Figure 6. Figure 7 shows the graphs comparing the explained variance against the number of Principal Components, using all the MNIST digits. As seen, the images generated using a smaller dimension of noise vector $Z_{\tau'}$ use fewer principal components to explain variance as compared to $Z_T$, $Z_\tau$.

We also provide a python script which implements the projection step to optimize the noise vectors in all three different cases mentioned in our main paper ($Z_T$, $Z_\tau$ and $Z_{\tau'}$). We used a generative model trained on MNIST images to perform the projection step, but the script can be utilized to do the same for any generative model.

### D.2   IGO for training Diffusion models

#### D.2.1   Generated Samples for CelebA dataset

The generated samples using the progressive generation method used by Ho et al. (2020) can be seen in Figure 8 and Figure 9. It can be observed from the samples in Figure 8, we utilize our intermediate layer route to sample images from a lesser starting corruption point compared to the final layer. Our code to incorporate the IGO can be found in our submitted code repository.

#### D.2.2   Implementation Details for MNIST dataset

To analyze the effect of $\alpha$, we perform another experiment, similar to Sec. 6.2 from the main paper, using MNIST dataset. During the forward pass, our simulated iterates $\mathbf{x}_t$ and $\tilde{\mathbf{x}}_\tau$ ($\tau = t/2$) are passed into the

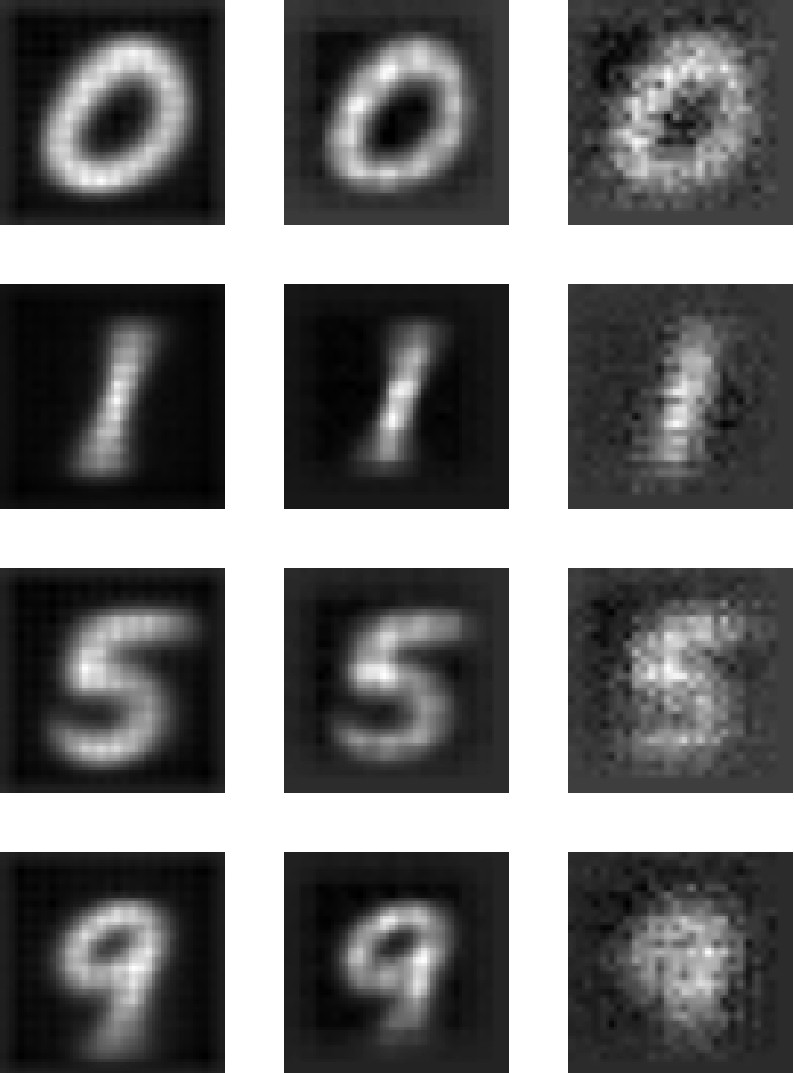

Figure 6: Reconstructed Images for Generative PCA with IGO. **First row:** from $Z_T$, **Second row:** from $Z_\tau$, **Third row:** from $Z_{\tau'}$

network, as shown in Figure 1. We use a convolution layer with a kernel of size 5 and stride 2 as the intermediate encoder $E_\tau$ and a deconvolution layer with a kernel of size 6 and stride 2 as the intermediate decoder $D_\tau$. The training is done using a convex combination of loss, the same as done in the case of CelebA dataset. Figure 10 summarises the PyTorch implementation of IGO in a U-net based Generative setting for the MNIST dataset.

### D.2.3 Generated Samples for MNIST dataset

For quantitative comparisons of the images generated from models using different values of $\alpha$, we utilized the PyTorch implementation of Fretchet Inception Distance (FID) Heusel et al. (2017) score provided by Seitzer (2020). The results can be seen in Figure 11. We can see that the values of $\alpha$ can be chosen depending on the trade-off between range expansion and the FID.

Figure 12 shows the Generated samples with the perturbation kernel, using Arnold's Cat Map as its drift function, for different values of $\alpha$. The generated samples do have some structural deformities, indicating that

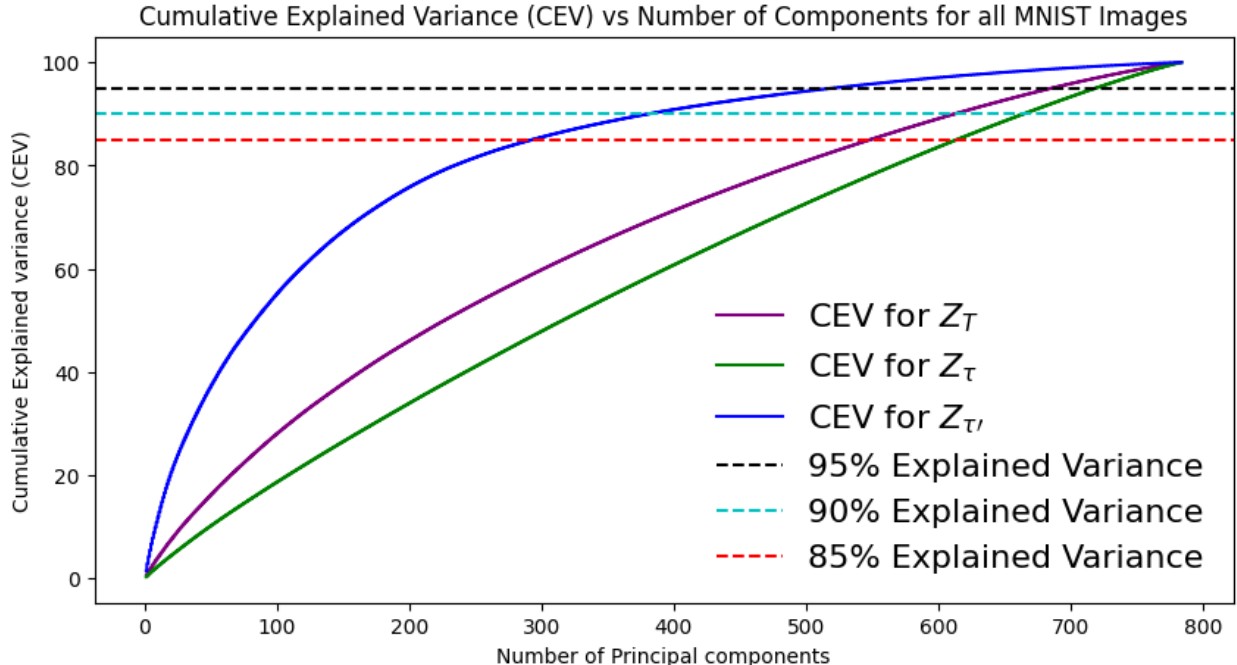

Figure 7: Number of Principal Components required for explaining variance.

the range of the generators is improved due to our setup. Moreover, note that our IGO scheme is robust with respect to $\alpha$ as shown by the gradual degradation in performance as we place more weight on the gradients provided by the intermediate iterates. Thus suggesting that using a combinatorial objective function can help increase the range of generative models.

### D.3   IGO on Panel Data

### D.3.1   Implementation Details

Experiments were run on NVIDIA 2080ti GPU for 300 epochs. First 100 epochs were used to train the encoder/decoder only, without any temporal component. For Rotating MNIST (2D data), the encoder-decoder model that we used in our experiments are described in Figure 13.

### D.4   IGO in 3D Processing

### D.4.1   Qualitative Results

Figure 14, 15 and 16 show the qualitative results from the experiment in Sec. 6.4, for different values of the hyperparameter $\alpha$.

### D.4.2   Quantitative Results

The comparison with the baselines for noise levels 1% and 2% are shown in Table 1 for the model trained with $\alpha = 0.8$. We can see that our model's performance is close to the state of the art model in the lower noise settings, while beating a lot of other baselines.

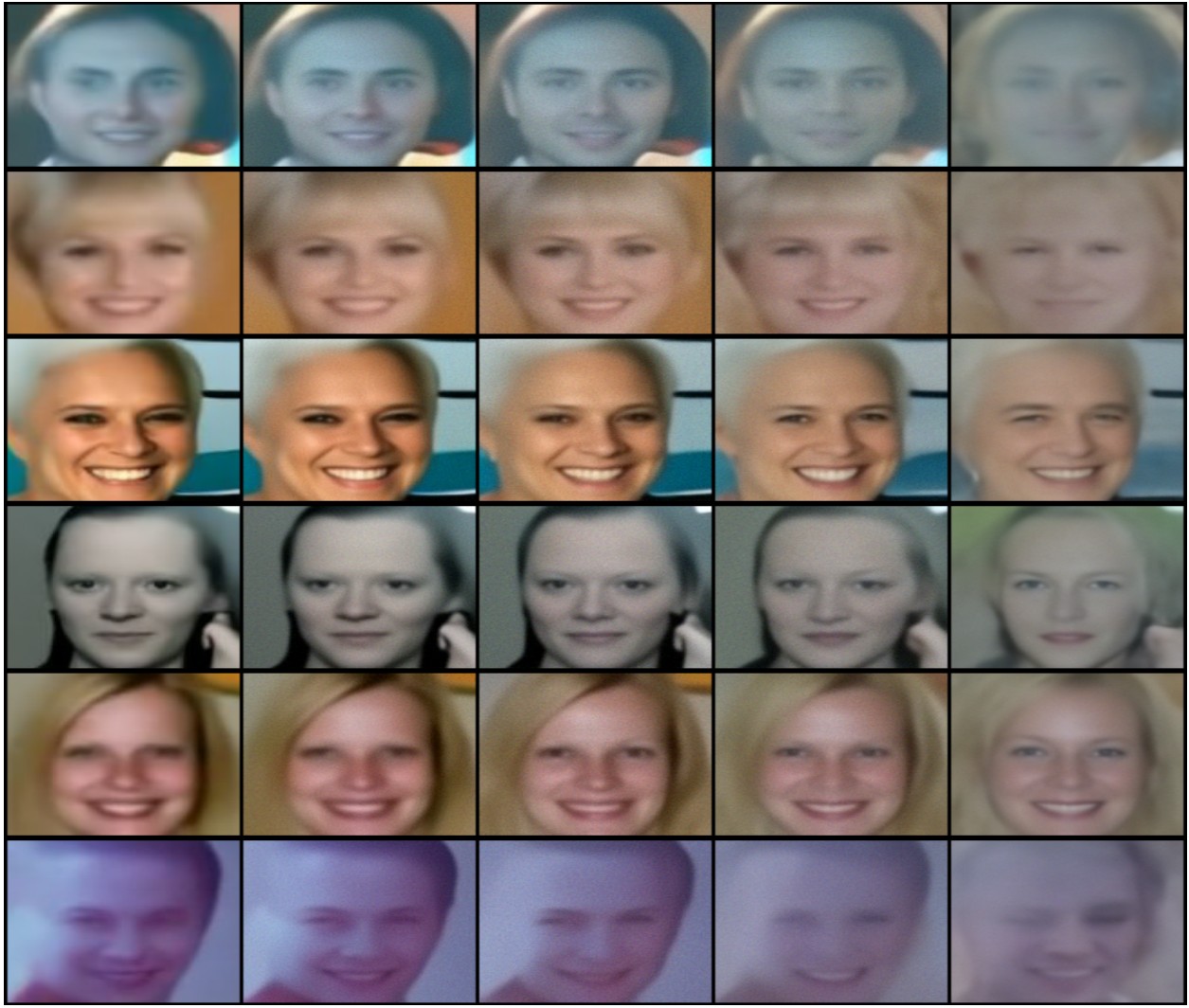

Figure 8: Samples generated from the Intermediate Layer (the right-most column being the closest to noise.)

| # Points | 10K (Sparse) | | | | 50K (Dense) | | | |
|---|---|---|---|---|---|---|---|---|
| Noise | 1% | | 2% | | 1% | | 2% | |
| Model | CD | P2M | CD | P2M | CD | P2M | CD | P2M |
| Bilateral Fleishman et al. (2003) | 3.646 | 1.342 | 5.007 | 2.018 | 0.877 | 0.234 | 2.376 | 1.389 |
| Jet Cazals & Pouget (2005) | 2.712 | 0.613 | 4.155 | 1.347 | 0.851 | 0.207 | 2.432 | 1.403 |
| MRPCA Mattei & Castrodad (2017) | 2.972 | 0.922 | 3.728 | 1.117 | **0.669** | **0.099** | 2.008 | 1.033 |
| GLR Zeng et al. (2020) | 2.959 | 1.052 | 3.773 | 1.306 | 0.696 | 0.161 | 1.587 | 0.830 |
| PCNet Rakotosaona et al. (2020) | 3.515 | 1.148 | 7.467 | 3.965 | 1.049 | 0.346 | 1.447 | 0.608 |
| DMR Luo & Hu (2020) | 4.482 | 1.722 | 4.982 | 2.115 | 1.162 | 0.469 | 1.566 | 0.800 |
| Score-Based PCD Luo & Hu (2021) | **2.521** | **0.463** | **3.686** | **1.074** | 0.716 | 0.150 | **1.288** | **0.566** |
| Ours ($\alpha = 0.8$) | 2.710 | 0.593 | 4.292 | 1.524 | 0.954 | 0.320 | 2.456 | 1.517 |

Table 1: Comparison against other denoising algorithms. The CD as well as the P2M scores are multiplied by $1e+4$.

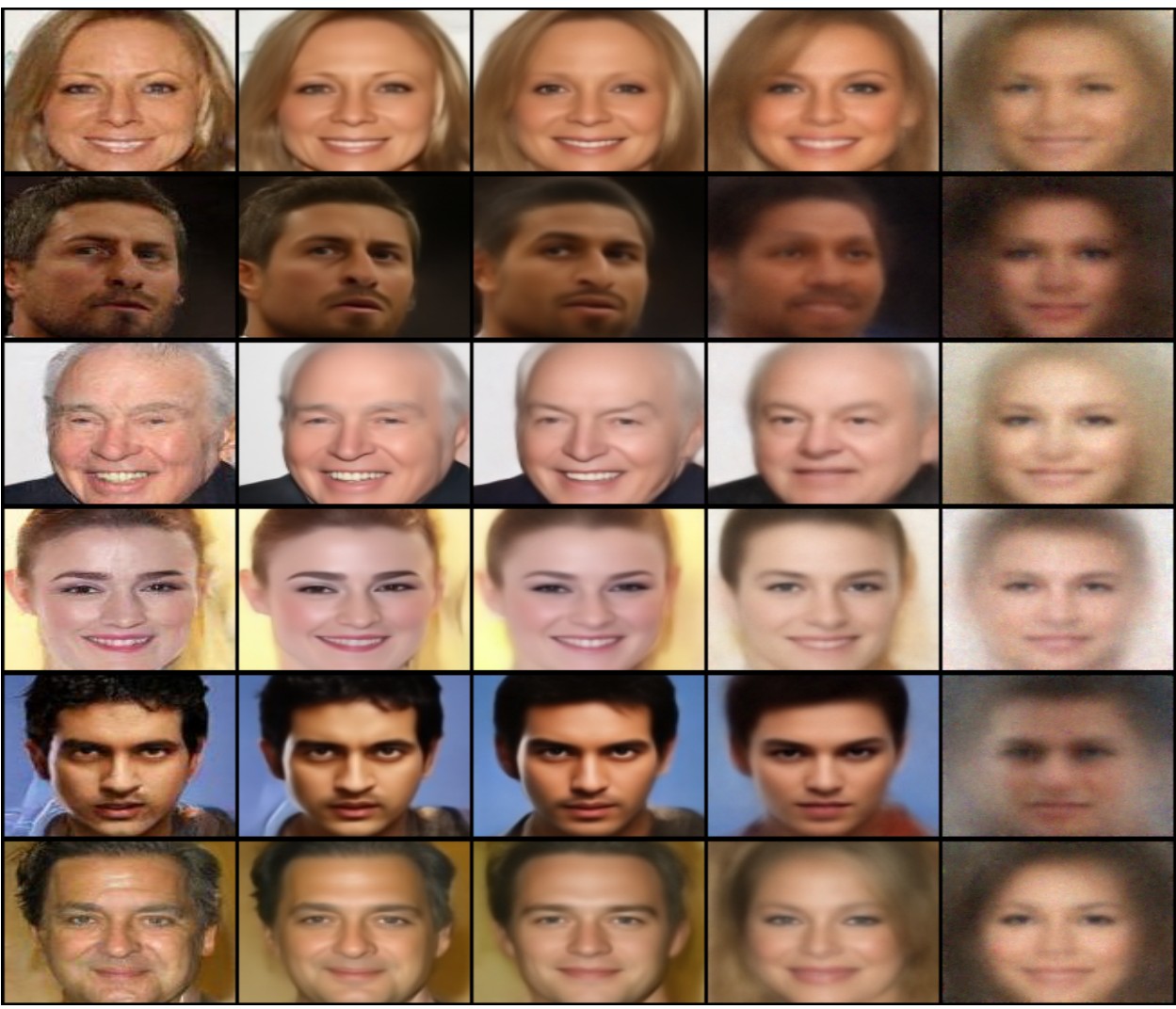

Figure 9: Samples generated from the Final Layer (the right-most column being the closest to noise.)

Encoder structure for IGO on MNIST

```
encoder = nn.Sequential(
        nn.Conv2d(1, 32, ks = (3,3),
                  stride=(1,1)),
        nn.Conv2d(32, 64, ks = (3,3),
                  stride=(2,2)),

        #Intermediate Encoder
        nn.Conv2d(1, 64, ks = (5,5),
         stride=(2,2)),

        nn.Conv2d(64, 128, ks = (3,3),
                  stride=(2,2)),

        nn.Conv2d(128, 256, ks = (3,3),
                  stride=(2,2))
        )
```

Decoder structure used for IGO on MNIST

```
decoder = nn.Sequential(
        nn.ConvTranspose2d(256, 128, ks = (3,3),
                  stride=(2,2)),
        nn.ConvTranspose2d(128, 64, ks = (3,3),
                  stride=(2,2)),

        #Intermediate Decoder
        nn.ConvTranspose2d(64, 1, ks = (6,6),
                  stride=(2,2)),

        nn.ConvTranspose2d(64, 32, ks = (3,3),
                  stride=(2,2)),

        nn.ConvTranspose2d(32, 1, ks = (3,3),
                  stride=(1,1))
        )
```

Figure 10: Implementing IGO on Pytorch involves encoding intermediate $\tilde{\mathbf{x}}$ with $E_\tau$, which eventually gets decoded by $D_\tau$.

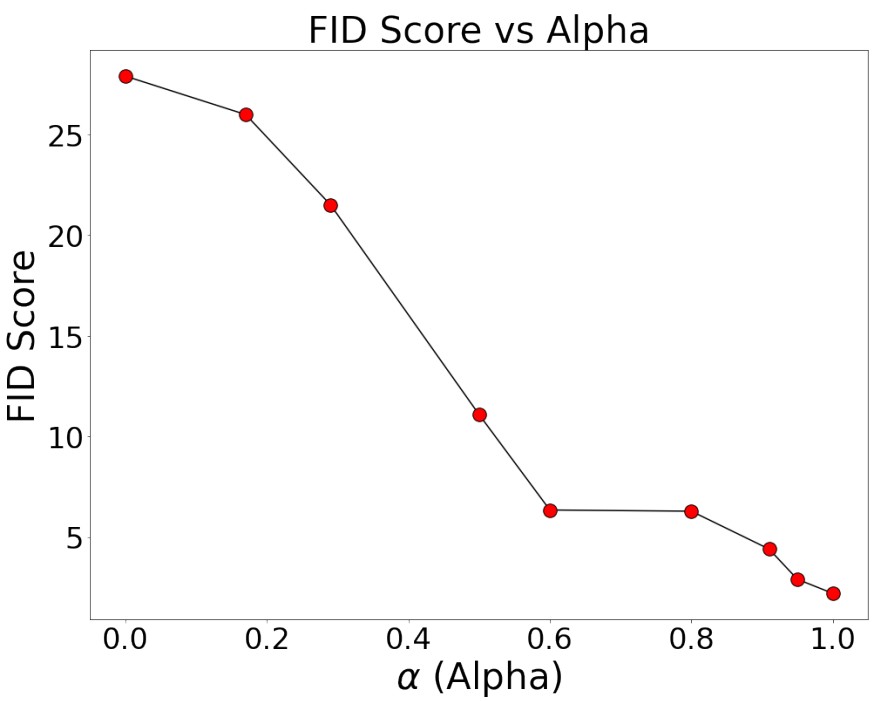

Figure 11: FID scores for Images generated using models trained using different values of $\alpha$

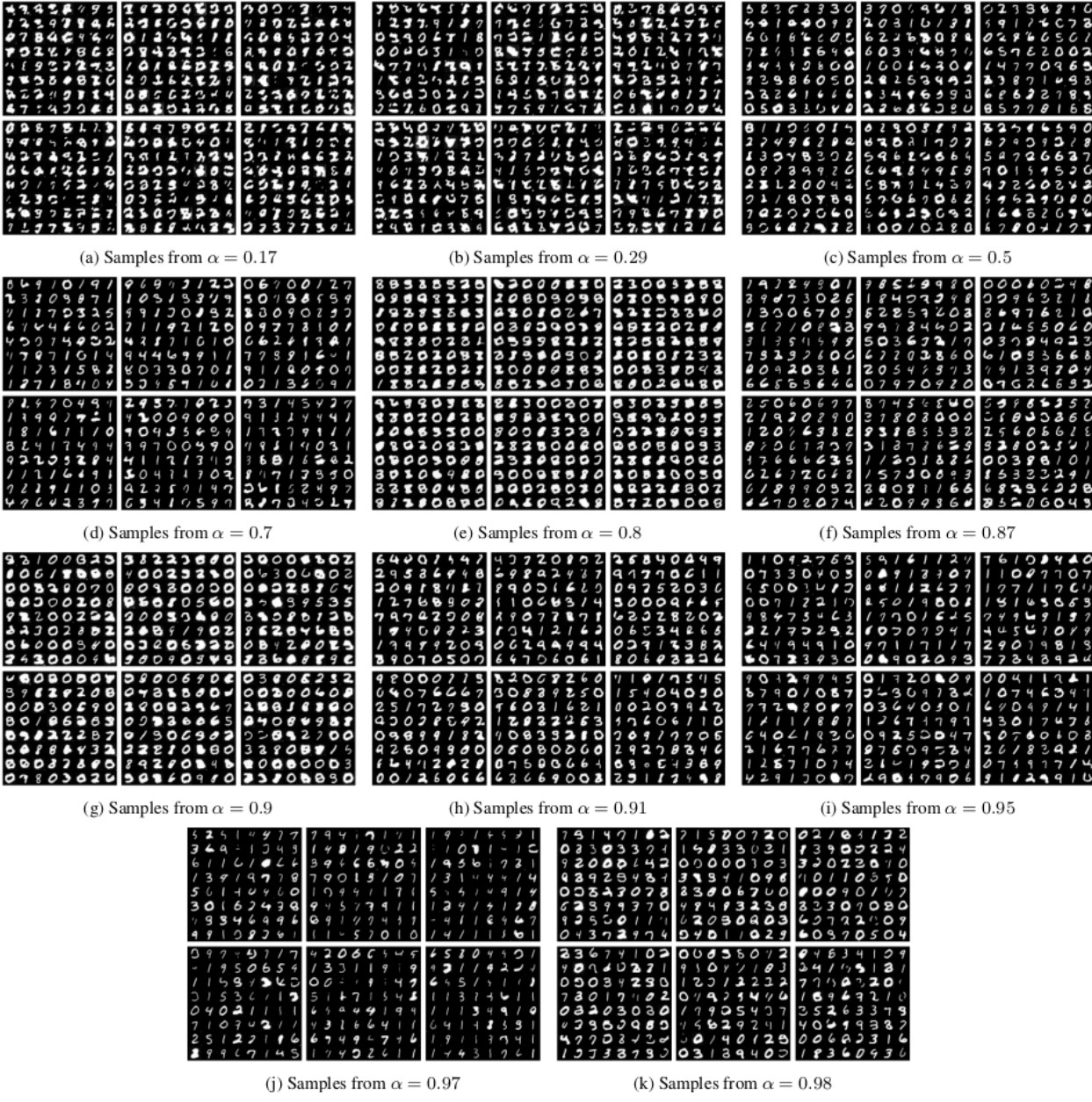

Figure 12: Samples for different values of $\alpha$

Encoder structure used in ROTATING MNIST

```
encoder = nn.Sequential(
        nn.Conv2d(input_dim, 12, ks,
                  stride=1, padding=1),
        nn.ReLU(),
        nn.Conv2d(12, 24, ks,
                  stride=2, padding=1),
        nn.ReLU(),
        nn.Conv2d(24, output_dim, ks,
                  stride=2, padding=1),
        nn.Flatten(2),
        nn.Linear(49, 1),
        nn.Flatten(1)
        )
```

Decoder structure used in ROTATING MNIST

```
extend_to_2d = nn.Linear(input_dim,
                         49 * input_dim)
decoder = nn.Sequential(
        nn.ConvTranspose2d(input_dim,
                           24,
                           ks,
                           stride=2,
                           padding=1,
                           output_padding=1),
        nn.ConvTranspose2d(24,
                           12,
                           ks,
                           stride=2,
                           padding=1,
                           output_padding=1),
        nn.ConvTranspose2d(12, output_dim, ks,
                           stride=1, padding=1),
        nn.Sigmoid(),
        )
```

Figure 13: Description of Encoder and Decoder used in experiment with 2D data structure: Rotating MNIST.

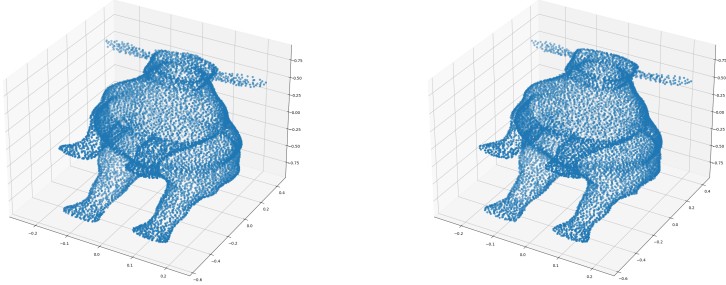

Figure 14: Denoised Image using a) Intermediate layer b) Final layer, $\alpha = 0.98$

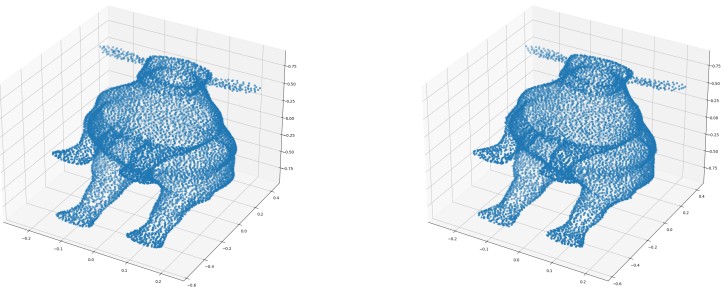

Figure 15: Denoised Image using a) Intermediate layer b) Final layer, $\alpha = 0.9$

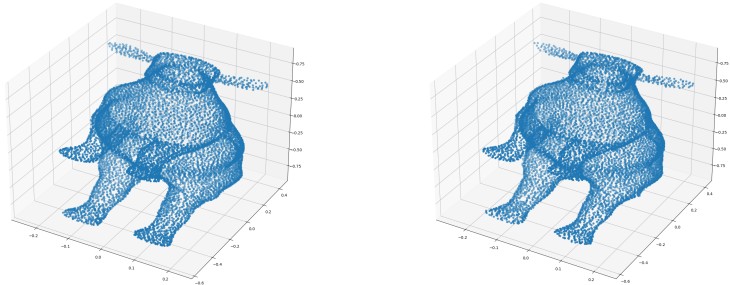

Figure 16: Denoised Image using a) Intermediate layer b) Final layer, $\alpha = 0.5$

