# OpenReview forum: "Using Intermediate Forward Iterates for Intermediate Generator Optimization"
_TMLR — Rejected by TMLR_

### Review · Reviewer_eHGg · 2024-07-22

**Summary Of Contributions:**

This paper introduces the Intermediate Generator Optimization (IGO), a new approach for score-based generative models. IGO leverages intermediate iterates of the forward process, enabling efficient training of score-based generative models under non-Gaussian forward processes. The experiments demonstrate that IGO can be utilized for multiple applications such as image generation and point cloud denoising.

**Audience:**

Yes

**Broader Impact Concerns:**

There are no broader impact concerns.

**Claims And Evidence:**

No

**Requested Changes:**

Please see the comments above.

**Strengths And Weaknesses:**

Strength
- The paper is well motivated.
- The paper shows the theoretical properties that the training of generative models with IGO offers good sample complexity.

Weakness
- The propose method assumes the availability of intermediate iterates, which is a strong assumption. Typically, any intermediate iterates are not given and only the final iterate $x_T$ is provided for generation, such as random Gaussian noise in Section 6.2 and a noisy point cloud data in Section 6.4. It seems that the experiments utilizing intermediate iterates are a kind of conditional generation, and thus the problem setting is different from standard unconditional generation. To emphasize practical usefulness, the paper should discuss the availability of intermediate iterates more thoroughly, and provide examples where intermediate iterates are naturally accessible.
- The diversity of samples generated by IGO is not clearly demonstrated. Although IGO utilizes multiple generators, the generator with intermediate iterates will produce samples related to those iterates. This suggests that IGO might generate less diverse samples. To address this, the paper should define sample diversity and provide a quantitative evaluation of the diversity achieved by IGO.
- The practical advantage of utilizing non-Gaussian corruption process is unclear.
- The analysis of training efficiency in Section 5 is confusing. The paper claims that the training $|T|$ DGM requires $O(P|T|)$ parameters, but as introduced in Section 6.2.2, the parameters of DGM are typically shared across time, suggesting the model only needs $O(P)$ parameters.
- The paper lacks comparison with other efficient methods for diffusion models. For example, Rombach et al. (2022) utilize the latent space like IGO, and Salimans et al. (2022) reduce the discretization size $T$ for efficient sampling. To demonstrate the efficiency advantage of IGO, it is desirable to compare IGO with these methods.


References
- Rombach et al. High-Resolution Image Synthesis With Latent Diffusion Models, CVPR, 2022.
- Salimans et al. Progressive Distillation for Fast Sampling of Diffusion Models, ICLR, 2022.

---

> ### Author Response · Authors · 2024-08-23
> **Clarification on the analysis of training efficiency in Section 5**
>
> We mean that if each time point uses a different set of model parameters then we need O(P|T|) where P is the number of parameters for a single time point, which may be infeasible in practice. Our IGO shares parameters so that the number of parameters is reduced to O(P), as discussed in Page 7. We have now clarified this in 6.2.2.

---

> > ### Author Response · Authors · 2024-08-23
> > **Diversity of samples generated by IGO**
> >
> > Some of our results shown in the paper and the supplementary show the quantitative and qualitative evaluation of the diversity achieved by using intermediate iterates/IGO. As part of section 6.4, figure 4- b and c show the diversity of our generators. On fine tuning the model, the cosine similarity between the two generators decreased by 10%. More quantitative results for IGO in 3D processing are provided in Table 1 of our supplementary.
> >
> > For quantitative comparisons of image generation, we have shown results using the MNIST dataset in Figure 11 of our supplementary. The figure shows the change of FID score as we gradually changed the value of alpha, i.e., alpha = 1 depicting no use of intermediate iterates and smaller alpha values depicting more and more use of the intermediate generators.
> >
> > Furthermore the generative PCA experiment results in Figure 7 of supplementary shows the graphs comparing the explained variance against the number of Principal Components. The image shows the difference between the explained variance when images were reconstructed using noise vectors from our different generators.

---

> > > ### Comment · Reviewer_eHGg · 2024-09-02
> > > **Response to the rebuttal**
> > >
> > > Thank you for your response. While I appreciate the efforts made to address my previous comments, I still have some remaining concerns.
> > >
> > > 1. Comment to "Clarification on the analysis of training efficiency in Section 5"
> > >
> > > - I understand that the number of parameters in IGO is O(P). However, what I mean is that popular existing methods also share parameters across time and do not use a different set of model parameters for each time point. Therefore, it seems that IGO does not offer advantage in terms of the number of parameters.
> > >
> > > 2. Comment to "Diversity of samples generated by IGO"
> > > - The author uses FID score as a diversity indicator. However, it's important to note that the FID score is also commonly used to evaluate sample quality. Consequently, high diversity scores may potentially be attributed to poor sample quality (i.e., low fidelity). The qualitative examples presented in the paper further reinforce these concerns.
> > > When assessing diversity in generative models, it's crucial to consider not only sample variation but also the coverage of real samples. To effectively demonstrate the high diversity of generative samples produced by IGO, the aspect of fidelity should not be overlooked or disregarded.
> > >
> > > 3 Comment to [the rebuttal](https://openreview.net/forum?id=QE9JKTrQ02&noteId=4KRzCR8xLf)
> > >
> > > - Handling real-world corruption patterns is definitely important. However, to fully demonstrate the strength of IGO from this perspective, the authors should acknowledge or discuss relevant prior work in this area. For example, Yang et al.[1] proposed a method to address real-world corruptions in diffusion models. The paper would benefit from mentioning and comparing to such existing approaches.
> > >
> > >     [1] Yang et al. real-world denoising via diffusion model, https://arxiv.org/pdf/2305.04457
> > >
> > > 4. Other comments
> > > - The availability of intermediate iterates x_{\tau} remains unclear. If x_{\tau} is generated from the final iterate x_{T}, then the generation by IGO essentially becomes a combination of one-step and multi-step generation from x_{T}. This approach would deviate from the concept of IGO as illustrated in Figure 1. If x_{\tau} is not generated from x_{T}, the method for acquiring x_{\tau} should be discussed.

---

### Review · Reviewer_fNgs · 2024-07-22

**Summary Of Contributions:**

This work proposes a novel way to train score-based models under non-Gaussian corruption procedures and incorporates physics-based information directly during training and/or inference. More specifically, the authors propose a new fully differentiable framework that explicitly models the corruption process within deep generative models, and which can be trained using standard backpropagation procedures. They analyze the statistical properties of the proposed framework (called Intermediate Generator Optimization (IGO)) using the recently introduced Intermediate Layer Optimization (ILO) framework for solving inverse problems (such as generative PCA). The authors also show applications of IGO on dense predictive tasks such as image extrapolation and point cloud denoising.

**Audience:**

Yes

**Claims And Evidence:**

No

**Requested Changes:**

See the Weaknesses above. Despite the intriguing algorithmic and experimental results presented in this work, the paper necessitates substantial rewriting prior to its potential acceptance.

**Strengths And Weaknesses:**

Strengths:
1. The paper presents a novel way to train score-based generative models under non-Gaussian corruption procedures, which is an innovative contribution to the field.
2. The authors provide a theoretical analysis of the IGO framework for downstream tasks, which helps to understand the properties and performance of the proposed method.
3. The paper demonstrates the effectiveness of the IGO framework through extensive experiments on various tasks, and it is good to see that the authors have included certain parts of the code in the appendices (and the code implementation in the supplementary material).

Weaknesses:
1. While the paper compares the performance of the IGO framework with some existing methods, the comparison may not be comprehensive enough. There may be other related methods that could have been included for a more thorough evaluation (e.g., the authors might compare with PPower for the generative PCA task).
2. I am familiar with both CSGM and score-based models, yet I still find this paper rather challenging to follow. Here are some examples that illustrate the issues in the writing:
- Please use consistent notations. For example, the authors use $\mathrm{d} w$ in Eq. (2), but use $d\bar{\mathbf{w}}$ in Eq. (3). The authors sometimes use $\mathbf{x}, \tilde{\mathbf{x}}_j, \mathbf{x}_t, \mathbf{x}(0), \mathbf{x}(T)$ (such as in Eqs. (2) and (4), and **Our Assumption on Forward Process** and **SDEs for Modeling Diversity**), but sometimes use $x_t, x(t), x(0)$ (such as in **Our Assumption on Forward Process** and **SDEs for Modeling Diversity**).
- In Eq. (4), it should be $\mathbf{z} \sim \mathcal{N}(\mathbf{0},\mathbf{I})$, instead of $z \sim \mathcal{N}(0,1)$ (and $\mathbf{z}$ should be $\mathbf{z}_j$). Additionally, $a(\tilde{\mathbf{x}}_j, t)$ should be $\mathbf{a}(\tilde{\mathbf{x}}_j, j)$ (or replace $j$ by $t_j$), and $b(\tilde{\mathbf{x}}_j, t)$ should be $b(\tilde{\mathbf{x}}_j, j)$.
- I am having difficulty understanding Eq. (5) (as well as Eqs. (6), (7), and (8)). I would appreciate it if the paper could explicitly specify what is meant by $L$. Additionally, compared to Eq. (1), I am confused about why there is no summation for $\mathbf{x}(0)$ in Eq. (5). Could this please be clarified in a more explicit manner?
- What are the **Input** of Algorithm 1?

---

> ### Author Response · Authors · 2024-08-23
> **Inconsistency in notations**
>
> Thank you so much for pointing out the inconsistencies in the equations that we missed. We have now corrected them in our revision.
>
> **Clarification on $d\bf{w}$ and $d\bar{\bf{w}}$** - Equation 2 depicts the forward SDE process, thus $d\bf{w}$ is a standard brownian motion. Whereas equation 3 is the reverse SDE equation & in this case $\bar{\bf{w}}$ was used to represent the brownian motion flowing backwards in time. We have now aligned them to avoid confusion. We have also made the changes to equation 4, as per your suggestion, and hopefully made it easier to understand.

---

> > ### Author Response · Authors · 2024-08-23
> > **Explanation of equations -**
> >
> > Training objective for Score based models, equation 1 in our paper:
> > $$
> > \theta^* = arg\min_\theta
> >    \{\mathbb{E}}\_t\left[\lambda(t) \{\mathbb{E}}\_{x(0)}\{\mathbb{E}}\_{x(t) \mid x(0) } \left[\||s_\theta(x(t), t) - \nabla_{x(t)}\log p_{t}(x(t) \mid x(0))\||_2^2 \right]\right]
> > $$
> >
> > Equation 5:
> >
> > $$
> > \min_{\theta} \sum_{\tau=t_1}^{t_T} \lambda\left(\tau\right) \sum_{i=1}^N \mathbb{E}\_{x_i(\tau) \mid x_i} L\left(s_{\theta}, x_i\left(\tau\right), \tau\right)
> > $$
> >
> > In equation 5, we discretized equation 1 from our paper, the standard score based training objective, into an empirical risk minimization (ERM) framework. Here, as you pointed out, the difference from equation 1 (reading from left to right) is: (a) the first expectation is time discretization, and (b) the second expectation is discretized over the data samples denoted by i, $
> > \sum_{i=1}^N \mathbb{E}_{x_i(\tau)|x_i}$
> >
> > This was done to simplify the two summations used in equation 1, as by summing over all samples i, we implicitly sum over the initial condition (x(0)) and also over $x_i(\tau)$|$x_i$.
> >
> > $L$ represents the loss function, usually the mean squared error, to measure how well the model approximates the true corrupted sample $x_i(\tau)$.
> >
> > Sorry about the typo in eq (5), we fixed $L\left(s_{\theta},x_i\left(\tau\right),t\right)$ to $L\left(s_{\theta},x_i\left(\tau\right),\tau\right)$. Thank you for pointing this out.
> >
> >
> > Equation 6:
> > $$\mathcal{R}(\tilde{x}\_{\tau},t)= L\left(D_{\tau}\circ s_{\tau} \circ E_{\tau},\tilde{x}\_{\tau},t\right)$$
> >
> > Then, by equation 6, we introduced a regularizer to handle and utilize the intermediate iterates in the training process. The purpose of this regularization function is to modify the parameters of the model (s) to effectively handle the intermediate iterates ($x_{\tau}$), where $\tau\in[0,1]$ sampled uniformly from $[0,1]$ and $\tau<t$. Note that $R$ uses the iterates only and not $\tau$, hence an intermediate generator. ($D_{\tau} \circ s_{\tau} \circ E_{\tau}$) represents the model used to handle intermediate iterates ($x_{\tau}$). This means that the input ($x_{\tau}$) is first processed by the encoder ($E_{\tau}$), the resulting latent representation is then processed by the score-based model restricted to that iterate ($s_{\tau}$), and finally, the decoder ($D_{\tau}$) reconstructs the data from its latent representation.
> >
> > Equation 7:
> >
> > $$\
> > \arg\min_{\theta,E_{\tau},D_{\tau}} \mathbb{E}\_t \left[\lambda(t)\mathbb{E}\_{x(0)}\mathbb{E}\_{(x(t)|x(0)}L(s_{\theta},x(t),t) + \lambda(\tau)\mathbb{E}\_{x(0)}\mathbb{E}\_{(x(\tau)|x(0)}L(D_{\tau}\circ s_{\tau,\theta}\circ E_{\tau},x(\tau),t)\right]
> > $$
> >
> > Equation 7 utilizes the regularization function from equation 6 to rewrite the score based training objective, specifically for IGO. This approach helps us use a convex combination of the scores or the losses at both the final time t and an intermediate time $\tau$ and allows the model to utilize information from different stages of the forward process.
> >
> >
> > Equation 8:
> > $$
> > R_{0}^T(\tilde{x}\)=\sum_{\tau=t_1}^{t_T} L\left(D_{\tau}\circ s_{\tau} \circ E_{\tau},\tilde{x}\_{\tau},T\right)
> > $$
> >
> > Finally equation 8 extends or generalizes the regularization function from equation 5 to handle multiple intermediate iterates, by summing the regularizer or the losses over all discretized time steps.

---

> > > ### Author Response · Authors · 2024-08-23
> > > **Inputs of Algorithm 1 -**
> > >
> > > We utilize Algorithm 1 to solve the PCA problem using our IGO model for the MNIST dataset, as part of our experiments in section 6.1. The algorithm uses the covariance matrix of the dataset ($V$), the initializing vector ($Z_i^{(0)}$​) and primary generative model’s weights at different time points $G_{T}$ and intermediate generator $G_{\tau}$, as its inputs. We chose to initialize ${Z}_i^{(0)}$​ using the largest diagonal entry of the covariance matrix, so it doesn’t change its values for the $i$ chosen. The algorithm is run for the intermediate time point $\tau$ and $T$. For our experiment we use it to optimize the noise vector for 3 different cases, as mentioned and now more clarified, in section 6.1.1. Our implementation of the algorithm is provided as part of our supplementary submission and can be found under the GenerativePCA subfolder.

---

> ### Comment · Reviewer_fNgs · 2024-09-06
> **Response about Inconsistency in notations**
>
> Thanks for the responses. I still have some concerns in the writing.
>
> - For "the authors use $\mathrm{d} w$ in Eq. (2), but use $d \bar{\mathbf{w}}$ in Eq. (3)", my point is that the authors should not arbitrarily use $d$ or \mathrm{d}, or use $w$ or \mathbf{w}. For the reverse process, of course you need to use the common notation such as $\bar{\mathbf{w}}$.
>
> - For the training objective in Eq. (1), the current version in the rebuttal is still confusing to me. Please specify what exactly is $L$ (and what are the parameters of $L$). I cannot see how $x_i(\tau)$ and $\tau$ can be set to be inputs of the standard objective function $L$. In addition, in Eq. (1), the authors should clearly specify the distribution of $t$.

---

### Review · Reviewer_zuQ5 · 2024-08-09

**Summary Of Contributions:**

This work investigates the problem of learning score-based generative models that can approximate the reverse process of a diffusion-like SDE that uses a non-affine drift function (in contrast to the affine drift function in a typical diffusion setup, which is required for the formulation of a closed form loss). Two candidate non-affine drifts studied are Lotka-Volterra and Arnold's Cat Map. The forward process is simulated using a discretized Euler approximation. A two-term loss function is proposed to learn such a score function. The first term uses a score-based loss given the original and approximate forward sample obtained from Euler discretization, and the second term uses a score-based loss given a (original, approximate forward pair) that uses half as many forward steps, and encoder and decoder nets appended before and after the base score network. Several experiments are conducted on generative PCA, image synthesis, trajectory prediction, and dense 3D prediction.

**Audience:**

No

**Claims And Evidence:**

No

**Requested Changes:**

I request that the paper undergoes a major revision to provide a clear and focused presentation of the impact of using non-affine diffusion as opposed to the standard affine diffusion. Experiments should clearly highlight differences and performance improvement. The text should be edited to provide less extraneous information and a clearer presentation of the motivation and impact. The use of the encoder and decoder needs to be established by an ablation study.

**Strengths And Weaknesses:**

Strengths:

* Studying methods to work with score-based models using non-affine forward drift functions is an interesting direction.
* The experimental results cover a wide range of situations.

Weaknesses:

* This paper lacked a clear motivation for studying non-affine score-based models. The experimental results provide very little comparison to the performance of typical affine score-based models. It seems like each experiment should include a comparison with a standard diffusion model as a baseline to show the impact of non-affine score-based models.
* Given the simplicity of affine transformation and the general desire to sample quickly from diffusion models, it seems like introducing non-affine drift could impact the ability of the model to generate samples in few steps along relatively straight pathways (see for example recent developments in Elucidated Diffusion Models or flow matching). This increases importance of motivating the reason for studying non-affine diffusion.
* The paper, including the text and experiments, generally lack focus. On a few occasions, the text provides too much detail about tangential topics (like the KDE discussion). The experiments do not highlight the significance of non-affine diffusion.
* The inclusion of the encoder and decoder does not seem intuitive, and the use of these modules is not ablated.

---

> ### Author Response · Authors · 2024-08-23
> **Lack of baseline and ablations -**
>
> We have included the baselines for all our experiments apart from the Generative PCA experiment, which showcases how the IGO framework can easily be used for downstream tasks like Generative PCA, by applying the PPower method.
>
>
> The IGO for training DGMs with non-Gaussian Drift section, which includes experiments on CelebA and MNIST (in supplementary) datasets, uses FID scores to compare the results to that from the paper Song et al. (2021). The results can be seen in section 6.2.2 for CelebA and Figure 11 in the supplement for MNIST, where alpha = 0 is the baseline setting from Song et al. (2021). As reported, for the CelebA dataset, the FID score of 3.5 obtained with the images from the intermediate layers is close to the FID score of 3.3 obtained with the usual layer images. Thus showing that the intermediate layers could be used to generate images of similar quality, and also with shorter pathways, so lesser computation.
>
> For IGO on Panel Data, we used Nazarovs, Jurijs, et al. as our baseline. We evaluated our model on MuJoCo Hopper and Rotation MNIST dataset on interpolation and extrapolation tasks and have shown the results in section 6.3.2. Our results showed that our pipeline using intermediate iterates outperformed the MSE’s of the ODE2VAE baseline for MuJoCo Hopper, and also the NODE & MEODE baselines for Rotation MNIST.
>
> For the 3D Processing case we have shown comparison to a large set of score-based 3D processing models, including Luo & Hu (2021). Due to the page constraints, additional results, using chamfer and point-to-mesh distance for comparison can be found in Table 1 of our supplementary. Although we didn’t outperform the top performing baseline of Luo & Hu (2021), our results were close to theirs for all noise levels.

---

> > ### Author Response · Authors · 2024-08-23
> > **The inclusion of the encoder and decoder does not seem intuitive, and the use of these modules is not ablated.**
> >
> > We use the same encoder and decoder models that are used in practice, for instance, as in U-Net architecture which uses skip connections also. In our final overall architecture, we allocate a specific portion of encoders and decoders for intermediate iterates. We conducted experiments to identify benefits of using the specific portion for intermediate iterates in Sections 6.2, 6.3 and 6.4. We are happy to perform more ablation, if needed.

---

### Author Response · Authors · 2024-08-23
**Rebuttal:**

We would like to thank all the reviewers for taking their time to go through our paper and provide insightful reviews. We have tried to clarify the technical concerns that were raised and restructured the paper to incorporate the suggestions.

Below we clarify some common concerns that reviewers expressed, about the lack of motivation behind studying non-gaussian corruption processes and their use in score-based models.

Motivation. When using iterative algorithms or multi-step noise conditioning processes to train neural networks, intermediate iterates are always generated, although sometimes implicitly. Using them to better aid these models by providing additional structure feels like a natural idea. For example in image extrapolation tasks using partially completed images as part of the extrapolation process is a variation of multi-step noise conditioning, which can be likened to the use of intermediate iterates.

Gaussian noise which are most popularly used in training diffusion models are very rarely corroborated in real-world noise and corruption patterns. The salt and pepper noises in images, artifacts in biological or medical images and the blurring/fogging of broadcast images are some examples of non-gaussian corruption. Using the non-gaussian noise as well as intermediate iterates in modeling such cases will help increase the robustness and accuracy of the model.

Lastly, non-affine processes also have many applications in physics based ML. Arnold's Cat map models rearrangement of configurations, so mass is conserved, whereas Lotka-Volterra is nonlinear, so some mass escapes the (local) imaging based measurements of physical processes. In this sense, we see a clear motivation to study various corruption processes from the perspective of downstream applications. Our work shows how to incorporate such physics information directly during training and/or inference, so we are quite excited about the future applications.

---

### Decision · Action_Editor_aD4F · 2024-09-19

**Recommendation:** Reject

**Comment:**

Although the paper tackles a relevant problem on an interesting topic, Reviewers zuQ5 and eHGg think the paper should be rejected. Reviewer fNgs leans towards acceptance, although the reviewer mentioned that "this submission needs significant polishment before official publication".

Overall, due to the limitations described on "Claims And Evidence", as well as the confusing technical aspects of the manuscript, I consider this paper requires additional work before it is ready for acceptance.

**Audience:**

The topic of this submission is score-based generative models, which is of interest for the community and in particular to researchers working on generative models. Unfortunately, due to the limitations described above, for the findings of this paper to be interesting to others more experiments and baselines would be needed.

**Claims And Evidence:**

This paper proposes Intermediate Generator Optimization (IGO), a method for training score-based generative models that overcomes the limitation of requiring closed-form corruption processes, being able to be incorporated into standard autoencoder architectures. The paper showcases IGO on image extrapolation and point cloud denoising tasks.

Unfortunately Reviewers zuQ5 and eHGg have concerns about the claims and evidence of the paper. In particular, Reviewer zuQ5 mentioned that there are not enough comparisons with SOTA that allow to draw meaningful conclusions (MNIST is too simple and Celeb-A does not provide evidence of the benefit of IGO). Reviewer eHGg shared concerns about IGO not offering an advantage (due to the way of counting model parameters), about the paper using FID to measure diversity (as oppoased to sample quality), and about the lack of discussion of relevant prior work. Finally, both Reviewers eHGg and fNgs point to several confusing technical details of the paper.

**Resubmission Of Major Revision:**

The authors may consider submitting a major revision at a later time.